# PriViT: Vision Transformers for Private Inference

**Naren Dhyani**                                                                  *nhd7682@nyu.edu*
*New York University*

**Jianqiao Cambridge Mo**                                                          *jm8782@nyu.edu*
*New York University*

**Patrick Yubeaton**                                                              *wpy2004@nyu.edu*
*New York University*

**Minsu Cho**                                                                   *chomd90@gmail.com*
*New York University*

**Ameya Joshi**                                                              *ameya.j005@gmail.com*
*New York University*

**Siddharth Garg**                                                                  *sg175@nyu.edu*
*New York University*

**Brandon Reagen**                                                                  *bjr5@nyu.edu*
*New York University*

**Chinmay Hegde**                                                             *chinmay.h@nyu.edu*
*New York University*

**Reviewed on OpenReview:** *https://openreview.net/forum?id=3CmPvcYJnm*

## Abstract

The Vision Transformer (ViT) architecture has emerged as the backbone of choice for state-of-the-art deep models for computer vision applications. However, ViTs are ill-suited for private inference using secure multi-party computation (MPC) protocols, due to the large number of non-polynomial operations (self-attention, feed-forward rectifiers, layer normalization). We develop PriViT, a gradient-based algorithm to selectively Taylorize nonlinearities in ViTs while maintaining their prediction accuracy. Our algorithm is conceptually very simple, easy to implement, and achieves improved performance over existing MPC-friendly transformer architectures in terms of the latency-accuracy Pareto frontier.

## 1 Introduction

**Motivation:** Deep machine learning models are increasingly being deployed by cloud-based providers, accessible only by API calls. In such cases, user data privacy becomes paramount, motivating the setting of private inference (PI) using secure multiparty computation (MPC). In its simplest form, MPC-based private inference is a two-party setup where a user (the first party) performs inference of their data on a model whose weights are owned by the cloud service provider (the second party), with both sides encrypting their inputs using cryptographic techniques prior to inference.

The main technical barrier to widespread deployment of MPC-based PI approaches is the *large number of nonlinear* operations present in a deep neural network model. Private execution of linear (or low-degree polynomial) operations can be made fast using cryptographic protocols like homomorphic encryption and/or secret sharing. However, private execution of nonlinear operations (such as ReLUs or softmax operations)

Table 1: *Comparison of various MPC-friendly approaches for deep image classification. NAS stands for neural architecture search; GD stands for gradient descent; CCT stands for Compact Convolutional Transformer. Our approach, PriViT, adaptively replaces various nonlinearities present in transformers with their Taylorized versions in order to reduce PI latency costs without drop in accuracy.*

| Approach | Arch | Methods | Units removed |
|---|---|---|---|
| Delphi (Mishra et al., 2020) | ConvNets | NAS + Poly Approx. | ReLU Layers |
| CryptoNAS (Ghodsi et al., 2020) | ResNets | NAS | ReLU Layers |
| Sphynx (Cho et al., 2021) | ResNets | NAS | ReLU Layers |
| DeepReDuce (Jha et al., 2021) | ResNets | Manual | ReLU layers |
| SNL (Cho et al., 2022) | ResNets | GD | Individual ReLUs |
| SENet (Kundu et al., 2023) | ResNets | GD | Individual ReLUs |
| MPCFormer (Li et al., 2022) | BERT | NAS + Poly Approx. | GELU Layers, Softmaxes |
| MPCViT (Zeng et al., 2022) | ViT | NAS + Poly Approx. | GELU layers, Softmaxes |
| SAL-ViT (Zhang et al., 2023) | CCT | NAS + Poly Approx. | Self-Attention Layers, Softmaxes |
| RNA-ViT (Chen et al., 2023) | CCT | Compressed Attention + Poly Approx. | Self-Attention Layers Softmaxes |
| **PriViT (this paper)** | ViT | GD + Poly Approx. | Individual GELUs, Softmaxes |

require Yao's Garbled Circuits, incurring high latency and storage overhead. Thus, unlocking fast, accurate, and efficient PI requires rethinking network design.

Consequently, an emerging line of work over the last five years has made several advances towards the design of "MPC-friendly" models; see more discussions below in Section 2. These methods approach PI from various angles. Approaches such as Delphi (Mishra et al., 2020) or Circa (Ghodsi et al., 2021) propose to replace ReLUs with MPC-friendly approximations, while approaches such as CryptoNAS (Ghodsi et al., 2020) and Sphynx (Cho et al., 2021) use neural architecture search (NAS) to search for network backbones with a minimal number of ReLUs. Peng et al. (2023) propose hardware-aware ReLU-reduced networks to achieve better latencies. Some of the latest approaches in this direction (SNL by Cho et al. (2022), and SENet by Kundu et al. (2023)) derive inspiration from network pruning.

However, the above body of work has gaps. The overwhelming majority of PI-aware model approaches have focused on convolutional architectures, and have largely ignored *transformer* models. In particular, the proper application of MPC to *vision transformer* (ViT) architectures remains less well-studied; see Table 1. Vision transformers (Dosovitskiy et al., 2020) currently list among the best performing deep models in numerous computer vision tasks, spanning image classification, generation, and understanding. On the other hand, vision transformers are very bulky, possessing an *enormous number* of nonlinear operations of different types: GELUs, softmaxes, and layer norms.

We are aware of three published approaches to addressing private inference for vision transformers. MPCViT (Zeng et al., 2022) proposes a MPC-aware NAS algorithm, along with a polynomial simplification of the attention mechanism coupled with knowledge distillation. SAL-ViT (Zhang et al., 2023) introduces a learnable approximation for the softmax and perform NAS to choose between two types of attention layers. RNA-ViT (Chen et al., 2023) proposes an attention architecture with a compressed attention map and a Taylor approximation of the softmax. The last two works assume hybrid ViT models, a different cryptographic security model, and have not made their models public, making direct apples-to-applies comparisons difficult; in this paper, we will focus on MPCViT, which (to us) is the gold standard for private inference over vision transformers.

**Our contributions and techniques.** We introduce **PriViT**, a very simple algorithm for designing MPC-friendly vision transformers. PriViT either improves upon (or is competitive with) MPCViT both in terms of latency and accuracy on TinyImagenet as well as CIFAR 10/100. Therefore, PriViT presents a

Table 2: *Accuracy-latency tradeoffs between PriVit and MPCViT. All latencies are calculated with the Secretflow (Ma et al., 2023) framework using the SEMI2k (Cramer et al., 2018) protocol. Detailed methodology is reported in Appendix A.* **Left:** *Comparison of PriVit versus MPCViT on TinyImagenet. PriViT achieves* **5.77× speedup** *for an isoaccuracy of approximately 64%.* **Right:** *Comparison of PriVit versus MPCViT on Cifar10. PriVit achieves* **1.14× speedup** *for an isoaccuracy of approximately 94%.* **Bottom:** *Comparison of PriVit versus MPCViT on Cifar100. PriViT achieves* **1.05× speedup** *for an isoaccuracy of approximately 78%.*

| PriViT | | MPCViT | | | PriViT | | MPCViT | |
|---|---|---|---|---|---|---|---|---|
| Acc | Latency (s) | Acc | Latency (s) | | Acc | Latency (s) | Acc | Latency (s) |
| 78.88 | 17.357 | 62.55 | 68.095 | | 96.58 | 7.935 | **94.3** | 7.273 |
| 78.16 | 15.303 | **63.7** | 53.992 | | 95.88 | 7.164 | 94.2 | 6.791 |
| 75.5 | 12.124 | 63.36 | 40.367 | | 95.58 | 6.789 | 94.1 | 6.606 |
| **64.46** | 9.350 | 62.62 | 25.996 | | **94.58** | 6.382 | 93.6 | 6.200 |

| PriViT | | MPCViT | |
|---|---|---|---|
| Acc | Latency (s) | Acc | Latency (s) |
| 80.6 | 7.316 | **77.8** | 7.099 |
| 79.78 | 7.12 | 76.9 | 6.782 |
| 79.27 | 6.989 | 76.9 | 6.590 |
| **78.49** | 6.782 | 76.4 | 6.228 |

new algorithmic approach for designing "privacy-friendly" vision transformers; see Table 2 for quantitative comparisons.

At a high level, our approach mirrors the *network linearization* strategy introduced in the SNL method by Cho et al. (2022). Let us start with a pre-trained ViT model with frozen weights, but now replace nonlinear operations with their *switched Taylorized* versions, i.e.,:

- replace each GELU unit, $\mathrm{GELU}(x_i)$, by:

$$c_i \mathrm{GELU}(x_i) + (1 - c_i)x_i \,,$$

- replace each row-wise softmax, $\mathrm{Softmax}(X_i)$ by:

$$s_i \mathrm{Softmax}(X_i) + (1 - s_i)\mathrm{SquaredAttn}(X_i),$$

where SquaredAttn is just the unnormalized quadratic kernel, and $c_i, s_i$ are *binary* switching variables. These switches decide whether to retain the nonlinear operation, or to replace it with its Taylor approximation (linear in the case of GELU, quadratic in the case of softmax[1]). Having defined this new network, we initialize all switch variables to 1, make both weights as well as switches trainable, and proceed with training using gradient descent.

Some care needs to be taken to make things work. We seek to eventually set most of the switching variables to zero since our goal is to replace most nonlinearities with linear units or low-degree polynomials; the surviving switches should be set to one. We achieve this by augmenting the standard cross-entropy training loss with a $\ell_1$-penalty term that promotes sparsity in the vector of all switch variables, apply a homotopy-style approach that gradually increases this penalty if sufficient sparsity is not reached, and finally binarize the variables via rounding. See Section 3.

**Implications.** We note that our point-of-comparison method, MPCViT, also follows a similar strategy as (Cho et al., 2022): judiciously replace both GELUs and softmax operations in vision transformers with their linear (or polynomial) approximations. However, they achieve this via a fairly complex MPC-aware NAS procedure. A major technical contribution of their work is the identification of a (combinatorial) search space, along with a differentiable objective to optimize over this space. Our PriViT algorithm is *conceptually much*

---

[1] Via several ablation studies we justify why we choose these particular approximations for these functions.

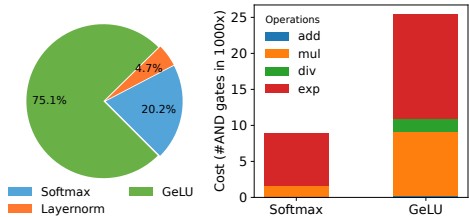

Figure 1: *Breakdown of latency in ViT-Tiny model of different non-linearities based on Delphi.*

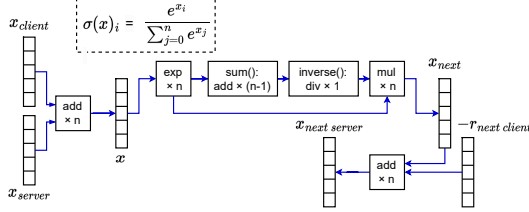

Figure 2: *Detailed steps of benchmarking the non-linearity cost for softmax.*

*simpler*, and can be applied out-of-the-box to any pre-trained ViT model. Moreover, we directly obtain a full family of model checkpoints during a single training run, at the cost of storing auxiliary mask variables. We commit to releasing all code and data needed to reproduce our results post-peer review.

While our focus in this paper is sharply on private inference, our results also may hold two broader implications for transformer architectures used in computer vision:

1. First, most nonlinear operations in transformers appear to be superfluous. PriViT is able to remove nearly 83% of GELUs and 97% softmax operations with less than 0.5% reduction in accuracy over CIFAR-100 (Krizhevsky et al., 2009).
2. Second, given a target overall budget of softmaxes and GELUs, PriViT overwhelmingly chooses to retain most of the nonlinearities in earlier layers, while discarding most of the later ones. These suggest that there is considerable room for designing better ViTs than merely stacking identical transformer blocks.

We leave a deeper investigations of these implications to future work.

## 2 Preliminaries

**Private inference.** Prior work on private inference (PI) have proposed methods that leverage existing cryptographic primitives for evaluating the output of deep networks. Cryptographic protocols can be categorized by choice of ciphertext computation used for linear and non-linear operations. Operations are computed using some combination of: (1) secret-sharing (SS) (Shamir, 1979; Micali et al., 1987); (2) partial homomorphic encryptions (PHE) (Gentry & Halevi, 2011), which allow limited ciphertext operations and (3) garbled circuits (GC) (Yao, 1982; 1986).

In this paper, our focus is exclusively on the Delphi protocol (Mishra et al., 2020) for private inference. We choose Delphi as a matter of convenience; the general trends discovered in our work hold regardless of the encryption protocol, and to validate this we measure latency of our PriViT-derived models using multiple protocols. Delphi assumes the threat model that both parties are honest-but-curious. Therefore, each party strictly follows the protocol, but may try to learn information about the other party's input based on protocol transcripts (Wang et al., 2022; Peng et al., 2023; Lu et al., 2021; Qin et al., 2022).

Delphi combines cryptographic primitives such as secret sharing (SS) and homomorphic encryptions (HE) for all linear operations, and garbled circuits (GC) for ReLU operations. For convolutional architectures, the authors of Delphi shows empirical evidence that ReLU computation requires 90% of the overall private inference time for typical deep networks. As a remedy, Delphi and SAFENET (Lou et al., 2021) propose neural architecture search (NAS) to selectively replace ReLUs with polynomial operations. CryptoNAS (Ghodsi et al., 2020), Sphynx (Cho et al., 2021) and DeepReDuce (Jha et al., 2021) design new ReLU efficient architectures by using macro-search NAS, micro-search NAS and multi-step optimization respectively.

Only recently has this line of literature turned its focus to transformer-style models. Approaches in this vein include MPCViT (Zeng et al., 2022), SAL-ViT (Zhang et al., 2023), and RNA-ViT Chen et al. (2023). MPC-ViT uses NAS techniques to find cheaper approximations of the softmax attention mechanism, while SAL-ViT and RNA-ViT propose new types of attention mechanisms. The latter two works claim state-of-the art over MPCViT; however, we found that a direct comparison is not possible for two reasons: (i) these

use compact convolutional transformers (CCTs), and (ii) they use the Crypten protocol for benchmarking MPC latency, which assumes the existence of a trusted third party (TTP). The lack of open source code or checkpoints for these two works makes it infeasible to port over their results to our comparison framework. Thus, our paper will focus on comparisons to MPCViT.

**Protocols for nonlinearities.** To standardize across different types of non-linear activations, we compare their Delphi (online) GC computation costs. We use the EMP Toolkit (Wang et al., 2016), a widely used GC framework, to generate GC circuits for nonlinear functions.

High-performance GC constructions implement AND and XOR gates, where XOR is implemented using FreeXOR (Kolesnikov & Schneider, 2008) and AND using Half-Gate (Zahur et al., 2015). With FreeXOR, all XOR gates are negligible, therefore we count the number of AND gates as the cost of each nonlinear function (Mo et al., 2023). To be consistent with prior work (Ghodsi et al., 2021), the activation functions also consider value recovery from Secret Sharing. Figure 1 (left) breaks down the GC cost of ViT for different nonlinearities, and (right) shows the # AND gates in Softmax and GeLU. Figure 2 breaks down softmax into fundamental operations, these operations are already synthesized and included in the EMP Toolkit library. Thus we simply add all the AND gates of these basic operations to arrive at the total number of AND gates of softmax operations.

**Setup.** Let $f_\mathbf{W} : \mathbb{R}^{n \times d} \to [0, 1]^C$ be a vision transformer that takes as input $n$ tokens (each of $d$ dimensions) and outputs a vector of probabilities for each of $C$ classes. Each of these tokens is a patch sampled from the original image, $\mathbf{X}$ and is indexed by $i$. As described, the transformer architecture consists of stacked layers of multi-headed self-attention blocks with nonlinearities like GeLU (Hendrycks & Gimpel, 2016) and Layernorm (Ba et al., 2016). ViTs use dot-product self-attention (see Equation 1) which additionally consists of $n$ row-wise softmax operations.

$$o = \frac{\text{Softmax}(\mathbf{X}\mathbf{W}_q\mathbf{W}_k^T\mathbf{X}^T)}{\sqrt{d}}\mathbf{X}\mathbf{W}_v. \tag{1}$$

To frame the computational challenges inherent to Vision Transformers (ViTs), we consider the ViT-base (12 layer) model designed for $224 \times 224$ images. Overall, this architecture consists of(approximately) $726,000$ GeLUs, $28,000$ softmax, and $4000$ layer norms. All the non-linearities, when viewed through the lens of the Delphi protocol, become extremely resource-intensive operations.

## 3 PriViT: MPC-Friendly ViTs

We propose to learn an architecture that circumvents these computationally heavy operations. Our approach surgically introduces appropriate Taylor approximations of the GeLU and softmax attention operations wherever possible (under the constraint that accuracy drops due to such approximations should be minimal.) The key challenge is to figure out where to perform these approximations.

### 3.1 Switched Taylorization

To begin, we focus on softmax and GeLUs and ignore layernorms; we found that these were far harder to Taylorize. For the former, we introduce auxiliary variables to act as switches. Given $f_\mathbf{W}$, let $\overline{C}$ and $\overline{S}$ be the total number of GeLUs and softmaxes. Further, let $\mathcal{S} = [s_1, s_2, ..., s_S]$ and $\mathcal{C} = [c_1, c_2, \ldots, c_G]$ be collections of binary switch variables defined for all instances of GeLU and softmax activations. Our goal here is to learn $\mathbf{W}, \mathcal{S}$, and $\mathcal{C}$ to ensure high accuracy with as few nonlinearities as possible. We also use $N$ to denote the number of tokens, $H$ to denote the number of heads and $m$ to denote the size of the token embedding (and consequently the output size of the feedforward MLP).

**GELUs.** In the case of GELU operations, we define a switched version of the GeLU activation:

$$f(c_i, \mathbf{x}_i) = c_i\text{GELU}(\mathbf{x}_i) + (1 - c_i)\mathbf{x}_i \tag{2}$$

$$\mathbf{y} = \big[f(c_1, x_1), f(c_2, x_2), \ldots, f(c_n, x_n)\big], \tag{3}$$

where $c_i$ is the corresponding auxiliary variable for the $i^{\text{th}}$ token, $\mathbf{x}_i$ is the $i^{\text{th}}$ input token embedding of dimension $m$ ($m$ being the MLP dimension) and $\mathbf{y} \in \mathbb{R}^{N \times m}$ is the output. During training, $c_i$ are initially real-valued, trainable, and are initialized to 1 at the start of training. During inference, we binarize all $c_i$ using an indicator function, $\mathbb{1}_{c_i > \epsilon}$, where $\epsilon$ is an appropriately chosen threshold. $c_i = 1$ implies that the GELU is preserved whereas $c_i = 0$ linearizes it.

**Softmax attention.** The next step is to reparameterize softmax attention. However unlike GELUs, choice of parameterization is not obvious here. As per the Delphi protocol, exponents are expensive in terms of latency; on the other hand, polynomials are comparatively cheaper. Division by a constant is preferable to division by an input-dependent parameter (like in softmax); therefore, we propose a modified 'Squared Attention' operation:

$$\text{SQUAREDATTN}(\mathbf{X}) = \frac{\left(\mathbf{X}\mathbf{W}_q\mathbf{W}_k^T\mathbf{X}^T\right)^2}{N}\mathbf{X}\mathbf{W}_v, \tag{4}$$

wherein we apply pointwise squaring instead of a row-wise softmax and divide by the number of tokens. Similar to our approach with GELUs, we further add a learnable auxiliary variable, $s_i$ for every row-wise softmax operation in the attention layer.

$$o = s_i\text{Softmax}(\mathbf{X}_i) + (1 - s_i)\text{SQUAREDATTN}(\mathbf{X}_i), \tag{5}$$

where $\mathbf{X}_i$ is the $i^{\text{th}}$ row of the attention matrix. As before, $s_i$s are initially real-valued, trainable and initialized to 1. The variables are binarized during inference allowing use of either Softmax or squared attention based on the values of $s_i$. Further ablations of different candidate attention functions are presented in the results sections.

## 3.2 Training PriViT models

To train PriViT models, we need to train three sets of variables: the weights of the transformer, $\mathbf{W}$, the switch variables for the GELU parameterization, $\mathcal{C}$, and the switch variables for the attention parametrization, $\mathcal{S}$. Our goal is to train a model that minimizes the number of nonlinearities to satisfy a given nonlinearity budget, that is, $\|\mathcal{C}\|_0 < C$, and $\|\mathcal{S}\|_0 < S$, while increasing the overall performance.

This is reminiscent of standard LASSO-style (Tibshirani, 1996) optimization. We therefore propose the following loss function to train the model,

$$L_{privit} = L(f_{\mathbf{W}}(\mathbf{X}), y) + \lambda_g \sum_{i=0}^{|\mathcal{G}|} |c_i| + \lambda_s \sum_{j=0}^{|\mathcal{S}|} |s_i|, \tag{6}$$

where $L$ is the standard cross-entropy loss. We then optimize for each of the variables until the required softmax attention and GELU budgets.

Optionally, we can also make use of knowledge distillation during both training and fine-tuning. We introduce a KL divergence loss on the soft labels generated by the teacher and student ViT model. This loss is added to the $L_{privit}$ loss defined in eq. 6. Thus our final minimization objective looks as follows,

$$\min_{\mathbf{W}, \mathcal{C}, \mathcal{S}} \quad L(f_{\mathbf{W}}(\mathbf{X}), y) + \lambda_g \sum_{i=0}^{|\mathcal{G}|} |c_i| + \lambda_s \sum_{j=0}^{|\mathcal{S}|} |s_i| + L_{kl}(f_{\mathbf{W}}(\mathbf{X}), f_{\mathbf{T}}(\mathbf{X})),$$

where $\mathbf{T}$ denotes the weights of the teacher model, and $L_{kl}$ is the KL divergence loss.

After every epoch, we count the number of GELUs and softmax attention operations by thresholding the $s_i$ and $c_i$ values. Once the model satisfies the required budgets, we freeze the chosen GELUs and softmax attention operations by binarizing all $s_i$ and $c_i$ values and fine-tune the model weights for the classification task. See Figure 15 for an overview of the method.

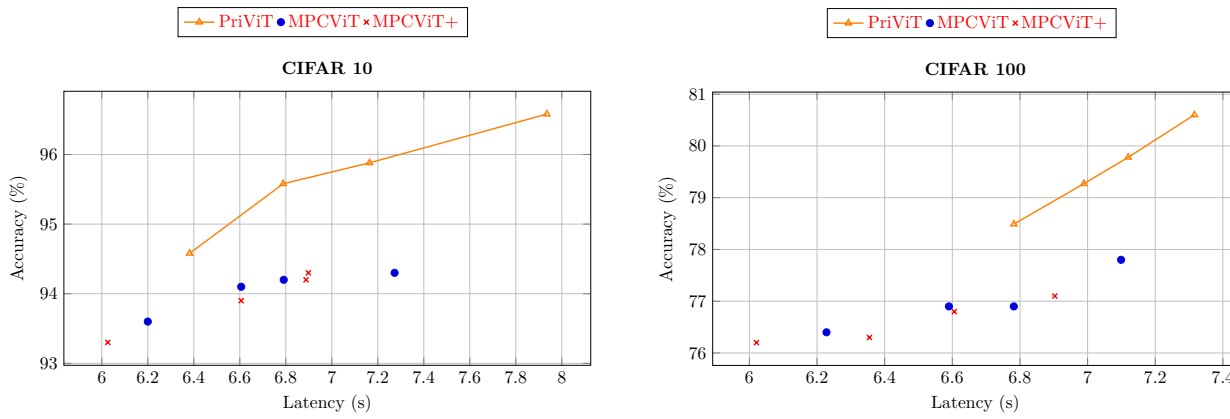

Figure 3: Comparison of PriViT over CIFAR 10/100 benchmarked against MPCViT, and MPCViT+.

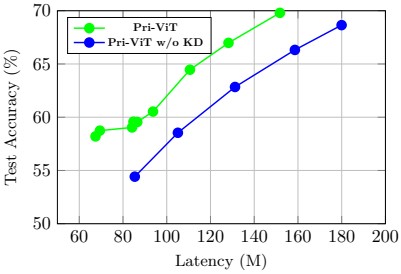

Figure 4: *We evaluated PriViT with and without KD. The x axis represents latency measured as per Section 4.1, while the y axis shows the accuracy on TinyImagenet. We observe an overall improvement in the latency-accuracy curve motivating the use of KD.*

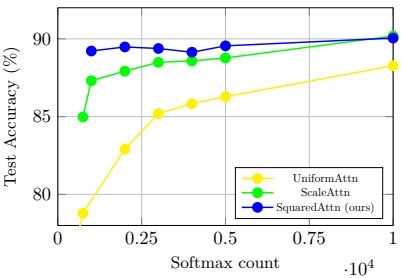

Figure 5: *We evaluated the PriViT algorithm using three attention operations: Uniform, Linear, and Squared Attention. The x-axis represents the target softmax count, while the y-axis shows the test accuracy on CIFAR-100.*

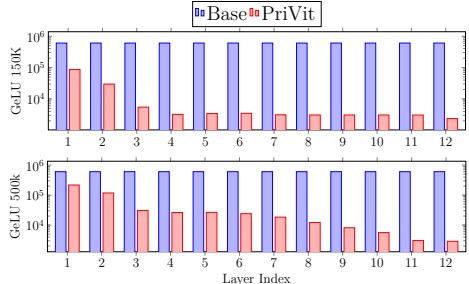

Figure 6: *Comparison of GELU distribution between ViT-base (Base) and PriViT without softmax linearization. The x-axis represents the model's layer index, while the y-axis shows log-scaled GELU operations per layer. With an input tensor size of $197 \times 3072$ for the GELU layer, each layer contains $197 \times 3072 = 605184$ GELU operations. **Top**: 150K target GELU. **Bottom**: 500K target GELU.*

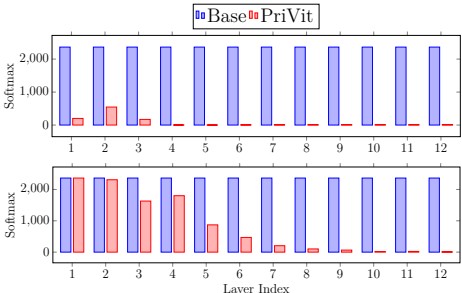

Figure 7: *Comparison of softmax distribution in ViT-base model (Base) versus PriViT without GeLU linearization. The x-axis denotes the layer index, while the y-axis shows the softmax operations per layer. With a $197 \times 197$ attention matrix across 12 heads, the ViT-base model totals 2364 softmax operations per layer. Notably, PriViT tends to substitute earlier layer softmaxes with linear operations. **Top**: 1K target softmax; **Bottom**: 10K target softmax.*

Table 3: *Latency comparison between PriViT and PriViT w/o Pretrain*

| PriViT | | PriViT w/o pretrain | |
|---|---|---|---|
| Latency (M) | Accuracy (%) | Latency (M) | Accuracy (%) |
| 271.59 | 75.5 | 234.18 | 53.57 |
| **151.74** | 69.8 | 194.78 | 54.66 |
| 128.23 | 66.98 | 167.20 | 55.59 |
| 93.71 | 60.53 | **153.31** | 55.92 |

## 4 Results

### 4.1 Experimental setup

**Architecture and datasets**. We apply PriViT algorithm to a pretrained checkpoint of ViT-Tiny (Steiner et al., 2021) that is trained on ImageNet-21k (14 million images, 21,843 classes) at resolution $224 \times 224$, and fine-tuned on ImageNet 2012 (1 million images, 1,000 classes) at resolution $224 \times 224$. The pretrained ViT Tiny checkpoints are made available by (WinKawaks, 2022). In this research work we focus on finetuning an existing model checkpoint like ViT Tiny on a target standard image classification dataset (CIFAR-10/100 (Krizhevsky et al., 2009) and Tiny-ImageNet). CIFAR-10/100 has images of size $32 \times 32$ while Tiny-ImageNet has $64 \times 64$. These images were resized to $224 \times 224$ before being given as an input. CIFAR-10 has 10 classes with 5000 training images and 1000 test images per class. CIFAR-100 has 100 classes with 500 training images and 100 test images per class. Tiny-ImageNet has 200 classes with 500 training images and 50 test images per class. We also perform hyperparameter tuning and present more details in Appendix A

**ViT teacher pretraining**. As the base model, we finetune a pretrained ViT-Tiny on CIFAR-10/100 for 10 epochs. We use AdamW (Loshchilov & Hutter, 2017) as the optimizer with an initial learning rate and weight decay as 0.0001 and 0.0001 respectively, and decay the learning rate after every 30 epochs by factor 0.1. We use the same hyperparameters for the TinyImagenet model.

**Joint optimization of student ViT and parametric non linearities**. We use Adam (Kingma & Ba, 2014) optimizer with learning rate equal to 0.0001. We use knowledge distillation and use soft labels generated by the teacher model with a temperature of 4. The total loss is then, $L = L_{\mathrm{PriViT}} + L_{\mathrm{KL}}$, where $L_{\mathrm{PriViT}}$ is Equation 6 and $L_{\mathrm{KL}}$ is the KL divergence loss between the logits of teacher and student model. The Lasso coefficient (Tibshirani, 1996) for parametric attention and GELU mask are set to $\lambda_g = 0.00003$ and $\lambda_s = 0.00003$ respectively at the beginning of the search. We set warmup epochs to 5 during which we don't change any hyperparameters of the model. Post warmup, we increment $\lambda_g$ by a multiplicative factor of 1.1 at the end of each epoch if the number of active GELUs of current epoch do not decrease by atleast 2 as compared to previous epoch. Note that a GELU/softmax is considered active if it's corresponding auxiliary variable is greater than threshold hyperparameter $\epsilon = 0.001$. We follow the same approach for $\lambda_s$, with a multiplicative factor of 1.1 and an active threshold of 200.

**Binarizing parametric nonlinearities, finetuning.** When the GELUs and softmax budgets are satisfied, we binarize and freeze the the GELU and softmax auxiliary variables. We subsequently finetune the model for 50 epochs using AdamW with a learning rate 0.0001, weight decay 0.0001 and a cosine annealing learning rate scheduler (Loshchilov & Hutter, 2016). Our finetuning approach continues to use knowledge distillation as before.

**Non-linearity cost comparison.** We conduct experiments to assess the computational cost of non-linear functions such as layernorm, softmax, and GeLU in comparison to ReLU within GC. The detailed results are reported in Table 12, brackets considers amortizing to a vector of inputs, e.g., a Layernorm(192) is an operation over a vector length of 192 is equivalent to $6504\times$ than the cost of a ReLU. It demonstrates that with a vector length of 197, all layernorm and softmax functions incur higher computational costs (i.e., number of ANDs) than ReLU. Specifically, they exhibit costs $6504\times$, $18586\times$ higher than that of ReLU respectively and for pointwise GELU, we saw a cost $270\times$ higher than that of ReLU. The cost of denominator of layernorm and

Table 4: *Comparison of training efficiency between PriViT and MPCViT on ViT-Tiny*

| | PriViT | | | MPCViT | | |
| Dataset | Latency | Accuracy | Epochs | Latency | Accuracy | Epochs |
|---|---|---|---|---|---|---|
| TinyImagenet | 151.75 | 69.8 | 293 | 381.42 | 62.55 | 600 |
| | 128.23 | 66.98 | 351 | 331.35 | 63.7 | 600 |
| | 110.60 | 64.46 | 342 | 307.45 | 63.36 | 600 |
| CIFAR 100 | 88.24 | 78.5 | 403 | 72.21 | 77.8 | 900 |
| | 75.92 | 77.74 | 447 | 71.77 | 76.9 | 900 |
| | 67.54 | 75.47 | 498 | 71.40 | 76.9 | 900 |

softmax can be amortized to the whole vector and thus incur less cost than GELU. We estimate the latency of each model generated by PriViT using these conversion factors. To showcase an example, we estimate the non-linearity cost of a hypothetical model with 1000 softmax operation, 1000 layernorm operations and 1000 GELUs, by taking the weighted sum of each operations with their corresponding latency factor.

**GELU replacement post training.** Mirroring the MPCViT+ approach, we also report the effect of PriViT with all GELUs replaced with ReLUs, we call this model PriViT-R, and the original PriViT model as PriViT-G. Such an optimization is effective For low GELU budgets as it introduces very minimal errors. For high GELU budgets, the error is quite significant that it affects the overall performance.

### 4.2 Comparisons on standard benchmarks

We benchmark PriViT against MPCViT, using the checkpoints publicly shared by the authors of Zeng et al. (2022). We calculate latencies with the Secretflow (Ma et al., 2023) framework using the SEMI2k (Cramer et al., 2018) protocol.

In our evaluation on various datasets, the performance of PriViT was benchmarked against both MPCViT and MPCViT+. We measure two metrics of importance – the latency (measured in seconds), and accuracy. An ideal model would achieve high accuracy with low latency.

1. **Tiny ImageNet**: In terms of the Pareto frontier on the Tiny ImageNet dataset, PriViT showcases notable improvement. On Tiny Imagenet, for an isoaccuracy of approximately 63% , PriViT achieved $5.77\times$ speedup compared to MPCViT as reported in Table 2.

2. **CIFAR-10**: Similarly, on the CIFAR-10 dataset, PriViT showcases noticeable improvements over MPCViT and MPCViT+. At an isoaccuracy of approximately 94%, PriViT achieves a $1.14\times$ and $1.08\times$ speedup over MPCViT and MPCViT+ respectively as reported in Table 2 and Table 9. This represents non-trivial speedup, *despite that fact that PriViT's model architecture is significantly larger than MPCViT's model architecture* as noted in Table 8. A graph of these values can be found in Figure 3.

3. **CIFAR-100**: Finally, on the CIFAR-100 dataset, PriViT also showcases noticeable improvements over MPCViT and MPCViT+. At an isoaccuracy of approximately 78%, PriViT achieves a $1.05\times$ and $1.02\times$ speedup over MPCViT and MPCViT+ respectively as reported in Table 2 and Table 9. We draw a similar conclusion to the analysis in CIFAR-10. PriViT achieves better accuracy with lower latency despite its larger model size. A graph of these values can be found in Figure 3.

### 4.3 Further comparisons with MPCViT

The majority of our numerical comparisons above are in relation to MPCViT, which is an important predecessor to our work. Our results above show that PriViT achieves a better Pareto frontier for private inference than MPCViT by achieving lower latencies at iso-accuracies across various datasets.

However, these comparisons are not completely apples-to-apples due to (i) differences in model size, and (ii) differences in pre-training. This is because MPCViT uses a smaller transformer architecture than the standard ViT backbone (which we use).

Therefore, we perform the following additional experiments:

1. We take the pre-trained ViT-Tiny checkpoint used in PriViT and apply NAS as described in the MPCViT paper.

2. We perform 300 epochs of NAS for each dataset in our comparisons (CIFAR-10/100 and Tiny-ImageNet). This is in line with the number of epochs reported in Zeng et al. (2022).

3. We then train this (post-NAS) ViT-Tiny on each benchmark dataset for varying values of $\mu$. The $\mu$ parameter represents the percentage of attention heads that use ReLU softmax instead of Scaled-Attention.

4. We train four models per dataset with $\mu$ values of 0.1, 0.3, 0.5, and 0.7 to match the training scheme of the MPCViT.

5. We were not able to replicate the knowledge distillation results present in Zeng et al. (2022). In our experiments found that the loss did not converge using their public code. Therefore, we opted to use a standard cross entropy loss function instead, and achieved reasonable accuracies with this method.

6. We trained the models for 300 epochs to match the training time of the MPCViT work. This leads us to a family of 12 models (four for each dataset). We call this "MPC-Tiny".

We present comparisons for PriViT with this new family of models in Tables 5, 6 and 7 below.

Table 5: MPC-Tiny (ie. MPCViT with the standard ViT backbone)

| $\mu$ | Dataset | Accuracy (%) | Latency (seconds) |
|---|---|---|---|
| 0.7 | CIFAR-10 | 91.48 | 42.03 |
| 0.5 | CIFAR-10 | 88.48 | 40.28 |
| 0.3 | CIFAR-10 | 93.97 | 36.80 |
| 0.1 | CIFAR-10 | 94.49 | 34.21 |
| 0.7 | CIFAR-100 | 40.22 | 43.17 |
| 0.5 | CIFAR-100 | 79.12 | 39.37 |
| 0.3 | CIFAR-100 | 75.04 | 36.82 |
| 0.1 | CIFAR-100 | 76.21 | 34.47 |
| 0.7 | Tiny Imagenet | 66.66 | 42.98 |
| 0.5 | Tiny Imagenet | 65.46 | 39.30 |
| 0.3 | Tiny Imagenet | 67.57 | 37.50 |
| 0.1 | Tiny Imagenet | 65.11 | 34.03 |

We now analyze our new results, starting with latency comparisons. We first see that all of the models present in MPC-Tiny are significantly slower than their counterparts in PriViT. Notably, the accuracy for the MPC-Tiny models is worse than both PriViT models and MPCViT models on average. This is likely due to the knowledge distillation procedure not transferring well to larger models. We also notice that for each value of $\mu$, the latency is similar regardless of the dataset, which is as expected and provides a good sanity check. In addition, we see that MPC-Tiny models have a relatively inflexible lower bound for latency. The $\mu$ parameter controls how much latency one can save by removing ReLU Softmaxes; even after we have reached $\mu = 0.1$, the latency is still very significant due to the large model size. This is in contrast to PriViT where we control the nonlinearity Taylorization at a scalar level (instead of attention head level) which enables us to push the latency down to significantly lower levels.

In conclusion, we show that a better comparison of PriViT can be made with MPC-Tiny as opposed to the original published MPCViT work. Our results show that PriViT has significantly lower latency at similar accuracies compared to MPC-Tiny.

Table 6: The original MPCViT (which uses a small transformer backbone)

| $\mu$ | Dataset | Accuracy (%) | Latency (seconds) |
|---|---|---|---|
| 0.7 | CIFAR-10 | 94.30 | 7.03 |
| 0.5 | CIFAR-10 | 94.20 | 6.66 |
| 0.3 | CIFAR-10 | 94.10 | 6.38 |
| 0.1 | CIFAR-10 | 93.60 | 6.06 |
| 0.7 | CIFAR-100 | 77.80 | 6.97 |
| 0.5 | CIFAR-100 | 76.90 | 6.62 |
| 0.3 | CIFAR-100 | 76.90 | 6.42 |
| 0.1 | CIFAR-100 | 76.40 | 6.06 |
| 0.7 | Tiny Imagenet | 62.55 | 69.64 |
| 0.5 | Tiny Imagenet | 63.70 | 55.45 |
| 0.3 | Tiny Imagenet | 63.36 | 40.86 |
| 0.1 | Tiny Imagenet | 62.62 | 26.76 |

Table 7: PriViT (which uses the standard ViT backbone)

| Dataset | Accuracy (%) | Latency (Seconds) |
|---|---|---|
| CIFAR-10 | 96.58 | 8.15 |
| CIFAR-10 | 95.88 | 7.28 |
| CIFAR-10 | 95.58 | 6.88 |
| CIFAR-10 | 94.58 | 6.42 |
| CIFAR-10 | 94.24 | 6.30 |
| CIFAR-10 | 93.60 | 5.91 |
| CIFAR-10 | 93.62 | 5.88 |
| CIFAR-100 | 80.60 | 7.30 |
| CIFAR-100 | 79.78 | 7.20 |
| CIFAR-100 | 79.27 | 7.04 |
| CIFAR-100 | 78.49 | 6.83 |
| Tiny Imagenet | 78.88 | 17.95 |
| Tiny Imagenet | 78.16 | 15.92 |
| Tiny Imagenet | 75.50 | 12.49 |
| Tiny Imagenet | 64.46 | 9.53 |

Table 8: *Base model architecture of PriViT and MPCViT*

| Model | Layers | Width | MLP | Heads | Image size | Patch size | params (M) |
|-------|--------|-------|-----|-------|------------|------------|------------|
| PriViT | 12 | 192 | 768 | 3 | 224×224 | 16×16 | 5.8 |
| MPCViT (Tiny Imagenet) | 9 | 192 | 384 | 12 | 64×64 | 4×4 | - |
| MPCViT (Cifar 10/100) | 7 | 256 | 512 | 4 | 32×32 | 4×4 | 3.72 |

Table 9: *Accuracy-latency tradeoff for MPCViT+. We compare this tradeoff with the PriVit tradeoff values from Table 2.* **Left:** *We first examine Cifar10 at an isoaccuracy of approximately 94%. PriVit achieves a* **1.08× speedup** *over MPCViT+.* **Right:** *We then examine Cifar100 at an isoaccuracy of approximately 78%. PriVit achieves a* **1.02× speedup** *over MPCViT+.*

| Cifar10 | | Cifar100 | |
|---------|-------------|----------|-------------|
| Acc | Latency (s) | Acc | Latency (s) |
| **94.3** | 6.898 | **77.1** | 6.903 |
| 94.2 | 6.888 | 76.8 | 6.606 |
| 93.9 | 6.606 | 76.3 | 6.355 |
| 93.3 | 6.026 | 76.2 | 6.021 |

## 4.4 Ablation studies

**Knowledge Distillation.** We incorporate knowledge distillation (KD) alongside supervised learning. To assess the contribution of KD to the overall performance, we trained PriViT on the TinyImagenet dataset with varying non-linearity budgets. We then compared its performance to a version of PriViT (as outlined in Figure 15) that does not employ a teacher model for knowledge distillation. Our results in Figure 4 indicate that, under identical latency conditions, incorporating KD enhances performance by approximately 5%.

**Contribution of pretraining.** In PriViT, we utilize a pretrained checkpoint, which is subsequently fine-tuned. Post fine-tuning, we introduce a parametric GeLU and attention mechanisms to decrease non-linearities in the model. To gauge the impact of using a pretrained model on the overall performance, we contrast the performance of PriViT with a variant of PriViT that is not built upon a pretrained model. Instead, this variant employs weights initialized from scratch and is trained with the same parametric non-linearity mask as used in PriViT to minimize non-linearities. The comparative outcomes of these approaches are presented in Table 3. Our findings reveal that, for comparable latencies, PriViT with the pretrained checkpoint outperforms the alternative by about 14%.

**Choice of softmax approximation.** We run PriViT over different softmax budget over CIFAR-100, and report the accuracy of the resulting model versus the number of original softmax attention retained. Lower number of softmax operations implies higher the number of softmax attention replaced with our candidate attention operation. As per Figure 5 we see almost no performance drop for SQUAREDATTN, roughly 5% drop in performance for SCALEATTN and 10% drop in performance for UNIFORMATTN in low budgets. Thus SquaredAttention outperformed the others across all softmax budgets, motivating its selection to replace the standard softmax attention in PriViT.

**Fine-grained versus layer-wise Taylorization** PriViT employs a unique approach where it selectively Taylorizes softmax and GELU operations. To probe the effectiveness of this method, we contrasted it with an alternative PriViT approach that Taylorizes a ViT model progressively, layer by layer. As illustrated in Table 10, our observations underscored the superiority of selective Taylorization.

**Visualization of non-linearity distribution.** To understand which nonlinearities are preserved, we investigate the distribution of PriViT models under different softmax and GELU budgets. From our observations in Figure 6 we can conclude that GELUs in earlier encoder layers are preferred over the ones in

Table 10: *Comparing PriViT and layerwise linearization of GeLU in a ViT model with 200k GeLUs. Six models were generated by replacing two GeLU layers at a time with Identity.*

| Layerwise GELU linearizing | | Pri-ViT | |
|---|---|---|---|
| Gelu (K) | Acc (%) | Gelu (K) | Acc (%) |
| 197 | 96.07 | 200 | 95.59 |
| 193 | 95.91 | 150 | 95.34 |
| 187 | 94.28 | 100 | 95.58 |
| 181 | 93.33 | 50 | 94.98 |
| 174 | 93.04 | 10 | 94.24 |
| 164 | 92.06 | 1 | 93.96 |
| 123 | 82.48 | | |
| 0 | 56.64 | | |

the later layers. From figure 7 we observe a similar trend in softmax distributions. We find this interesting, since the trends reported in earlier work on convolutional networks are in the reverse direction: earlier layers tend to have a larger number of linearized units. Understanding this discrepancy is an interesting question for future work.

Table 4 shows that PriViT, at a similar accuracy ( 64%), requires about half the training epochs compared to MPCViT on TinyImagenet. For an isolatency of 75M on CIFAR 100, PriViT also needs only about 50% of MPCViT's training epochs. This demonstrates PriViT's enhanced efficiency and scalability, making it a promising alternative to MPCViT, particularly in situations valuing efficiency and performance.

## 5 Conclusion

We introduce PriViT, a new algorithm for designing MPC-friendly vision transformers, and showed its competitive performance on several image classification benchmarks. A natural direction of future work is to extend similar techniques for designing other families of transformer architectures, such as Swin Transformers, as well as encoder-decoder transformer architectures. A key limitation of PriViT is the difficulty in properly Taylorizing layer norms without introducing instability in the training; we leave an in-depth study of this question to future work.

## Impact Statement

This paper aims to improve the efficiency of private inference for vision transformer architectures. Facilities who wish to perform inference on vision models while keeping user data private (such as hospitals) would gain the ability to use third party pretrained models without risking the leakage of confidential data. However, this advancement could also open the door to malicious users who would be able to use models without revealing their intentions. Considering both perspectives, we believe that the pros of advancing this field vastly outweigh the cons and thus believe it will lead to a net-positive impact.

### Acknowledgments

This work was supported in part by the US Department of Education's GAANN (Graduate Assistance in Areas of National Need) Fellowship Program. The research was also supported in part by the NSF CAREER award #2340137, NSF award #2154119 and by DARPA under the Data Protection in Virtual Environments (DPRIVE) program, contract HR0011-21-9-0003.

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

Table 11: *Accuracy-latency tradeoffs for PriViT applied to DeiT-Tiny.*

| Cifar10 | | Cifar100 | | Tiny Imagenet | |
|---|---|---|---|---|---|
| Acc | Latency (s) | Acc | Latency (s) | Acc | Latency (s) |
| 97.12 | 29.121 | 85.71 | 28.753 | 78.17 | 28.711 |
| 96.92 | 11.494 | 85.44 | 15.036 | 78.11 | 25.270 |
| 96.81 | 10.205 | 84.12 | 10.260 | 77.48 | 17.586 |
| 96.28 | 9.032 | 82.17 | 8.339 | 76.84 | 13.062 |
| 84.73 | 5.883 | 78.92 | 6.723 | 75.68 | 11.481 |

## A   Supplementary results

**PriViT on different size architectures.** Figure 8 provides a comparison of PriViT over different ViT variants from 5M (ViT-Tiny) to 80M (ViT-Base) parameters. We see that PriViT is able to produce a similar latency accuracy trade off across different model sizes emphasizing it's generalizability.

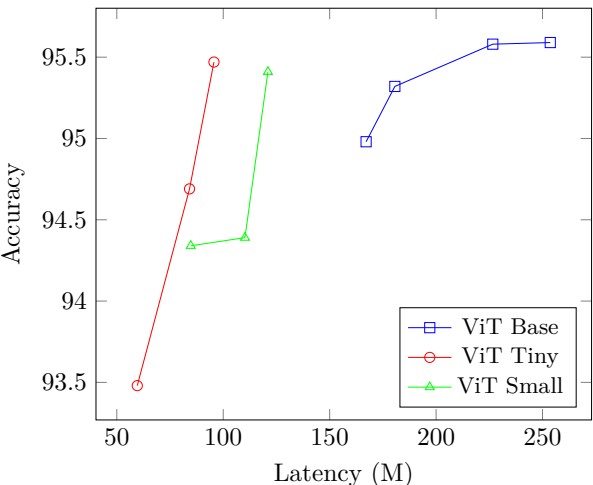

Figure 8: PriViT produces similar latency accuracy trade off for a variety of ViT models over CIFAR 10, highlighting it's scalability to large model sizes

**PriViT on different ViT architectures.** We study PriViT on the Data-Efficient Image Transformer (DeiT) (Touvron et al., 2021) to see how applicable the algorithm is to other ViT style architectures. We perform latency and accuracy tests for DeiT-Tiny on CIFAR-10, CIFAR-100, and Tiny Imagenet. Results are found in Table 11. The accuracy-latency tradeoff is very similar in the DeiT-Tiny model to the ViT-Tiny model. This suggests that PriVit is applicable to ViT style architectures as a whole, and not only to the original ViT architecture.

**Analysis of performance degradation.** In this analysis, we aim to compare the performance of trained PriViT models with their finetuned versions. Our analysis is based on the class-level accuracy metric from the Tiny ImageNet dataset, which consists of 200 classes. We focus on three specific parameters to understand the performance degradation:

*Maximum Difference in Accuracy:* We assess the greatest disparity in accuracy across all 200 classes between the PriViT and finetuned models.

*Overall Accuracy Difference:* We compute the average accuracy difference between the finetuned and the PriViT models across all 200 classes.

Table 12: *Non-linearity cost normalized to the cost of one ReluOp which is 1 ReLU operation over a scalar value.*

| PriViT | | MPCViT TinyImagenet | | MPCViT CIFAR-10/100 | |
|---|---|---|---|---|---|
| Function | # ReluOps | Function | # ReluOps | Function | # ReluOps |
| Softmax(197) | 18586 | ReLU Softmax(257) | 4428 | ReLU Softmax(65) | 1133 |
| Layernorm(192) | 6504 | Layernorm(192) | 6504 | Layernorm(256) | 8614 |
| GeLU(1) | 270 | GeLU(1) | 270 | GeLU(1) | 270 |
| $x^2$(197) | 3248 | | | | |

Table 13: Performance degradation of PriViT models compared to finetuned model on tinyimagenet.

| Accuracy | Latency (M) | Max Difference | Mean Difference | Variance ($10 \times 10^{-3}$) |
|---|---|---|---|---|
| 69.8 | 151.75 | 30.00% | 1.85% | 6.8 |
| 66.98 | 128.23 | 34.00% | 4.68% | 6.9 |
| 64.46 | 110.60 | 34.00% | 7.21% | 8 |
| 60.53 | 93.72 | 40.00% | 11.13% | 9 |
| 59.55 | 86.51 | 34.00% | 12.12% | 9.4 |
| 59.58 | 84.78 | 36.00% | 12.08% | 9.8 |
| 59.04 | 84.13 | 36.00% | 12.63% | 9.9 |
| 58.74 | 69.42 | 40.00% | 12.92% | 9.4 |
| 58.2 | 67.43 | 36.00% | 13.48% | 8.7 |

*Variance in Accuracy Difference:* We analyze the consistency of the differences in accuracy across the 200 classes by calculating the variance.

Table 13 highlights that average accuracy degradation is anywhere between 1-13% for different non-linearity budgets but certain classes seem to be more adversely affected even in low budgets as the max class level difference in accuracy is consistent around 30%.

**Hyperparameter Tuning.** Following (Hassani et al., 2021) we use CutMix (Yun et al., 2019), Mixup (Zhang et al., 2017), Randaugment (Cubuk et al., 2020), and Random Erasing (Zhong et al., 2020) as data augmentation strategy. We probed multiple hyperparameter strategies for the joint optimization phase of PriViT to ensure consistent good performance over multiple configurations of non-linearity budgets of softmax and GELUs. Specifically we describe these strategies as follows:

**Late-Binarized Epoch (Strategy 1)**: This strategy involved 10 post-linearization training epochs. The binarization of auxiliary parameters, *s* and *c*, occurred late in the process, specifically after the linearization was complete. The penalty increment condition for this method was checked when the reduction in the softmax and GELU coefficients per epoch was less than 200 and 2, respectively. Both masks began with identical penalties, signifying an 'equal' starting penalty.

**Late-Binarized Incremental (Strategy 2)**: This strategy also encompassed 10 training epochs with late binarization. Here, the penalty increment condition was activated with an increase in the softmax and GELU coefficients per epoch. The starting penalty for both masks was 'equal'.

**Late-Binarized Divergent Penalty (Strategy 3)**: Much like Strategy 2, this involved 10 epochs with late binarization and an increment condition based on softmax and GELU coefficient rises. However, the initial penalty was set to 'unequal', making the softmax penalty 20 times higher than the GELU penalty.

**Early-Binarized Incremental (Strategy 4)**: This strategy shared several similarities with Strategy 2, including 10 training epochs and an increment condition based on coefficient increases. The difference, however, lay in its early binarization, occurring during the freezing of the auxiliary parameters. The starting penalty was kept 'equal' for both masks.

Table 14: We test the different finetuning strategies described in  A. We run PriViT for different softmax and GELU budgets and compare the test accuracy of the resulting model over CIFAR-100. We observe that strategy 5 works the best across a wide range of target softmax and GELU budgets.

| # Softmax (K) | # Gelu (K) | Strategy 1 (Acc. %) | Strategy 2 (Acc. %) | Strategy 3 (Acc. %) | Strategy 4 (Acc. %) | Strategy 5 (Acc. %) |
|---|---|---|---|---|---|---|
| 10 | 5 | 77.68 | 76.74 | - | 77.82 | **78.83** |
| 5 | 5 | 76.27 | 75.99 | 75.72 | - | **77.63** |
| 5 | 1 | 76.73 | 75.21 | 76.24 | - | **77.08** |
| 2 | 10 | 76.04 | 75.23 | - | 74.65 | **76.35** |
| 2 | 1 | 75.92 | 74.84 | 76.45 | - | **76.97** |
| 1 | 5 | 76.12 | 74.99 | 76.32 | - | **76.96** |

**Prolonged Early-Binarized Epoch (Strategy 5)**: Spanning 50 post-linearization training epochs, this strategy adopted an early binarization approach. The penalty increment condition was activated when the reduction in softmax and GELU coefficients per epoch was under 200 and 2, respectively. The masks were initialized with 'equal' penalties.

Each of these strategies offered unique configurations in terms of epoch durations, binarization timings, increment conditions, and starting penalties, enabling a comprehensive assessment of the PriViT algorithm's performance under various conditions.

We test the different finetuning strategies described here by taylorizing PriViT for different softmax and GELU budgets and compare the test accuracy of the resulting model over CIFAR-100. Table 14 highlights the comparative performance of all the strategies that we described. Strategy 5 seems to be performing best over different configuration of nonlinearity budget which is important as we would want to find the best model peformance for a particular non-linearity budget.

**Grid search of softmax and GELU configuration.** In order to elucidate the nuanced trade-off between softmax and GeLU operations, we executed a systematic grid search across an extensive parameter space encompassing varied softmax and GeLU configurations. Upon analysis of models exhibiting iso-latencies, as demarcated by the red lines in figure 9, it became evident that the trade-off dynamics are non-trivial. Specifically, configurations with augmented softmax values occasionally demonstrated enhanced performance metrics, whereas in other scenarios, models optimized with increased GeLU counts exhibited superior benchmark results.

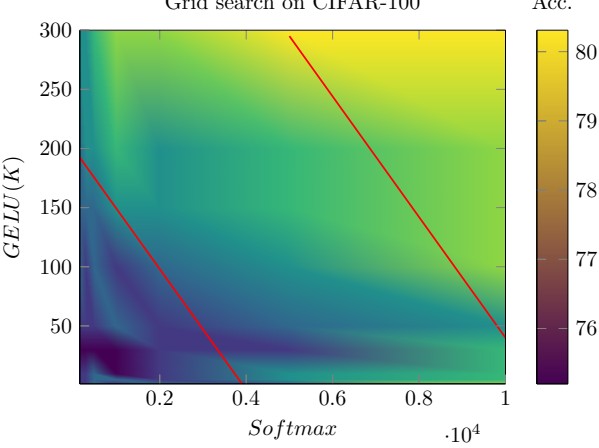

Figure 9: PriViT produces a Pareto surface mapping the tradeoff between GeLU and softmax budgets over CIFAR-100.

**Taylorizing only one type of non-linearity.** The PriViT algorithm's standout capability is its simultaneous linearization of GELU and softmax operations, enabling a myriad of model configurations. In our focused experiment, we exclusively linearized GELU operations and anchored the auxiliary softmax parameter $S$, binarizing it to activate only the SoftmaxAttention mechanism. Despite extensive GELU substitutions, as reported in 10 the PriViT model displayed notable resilience on CIFAR-10 and CIFAR-100 datasets, with only slight performance drops, underscoring its robustness in varied setups.

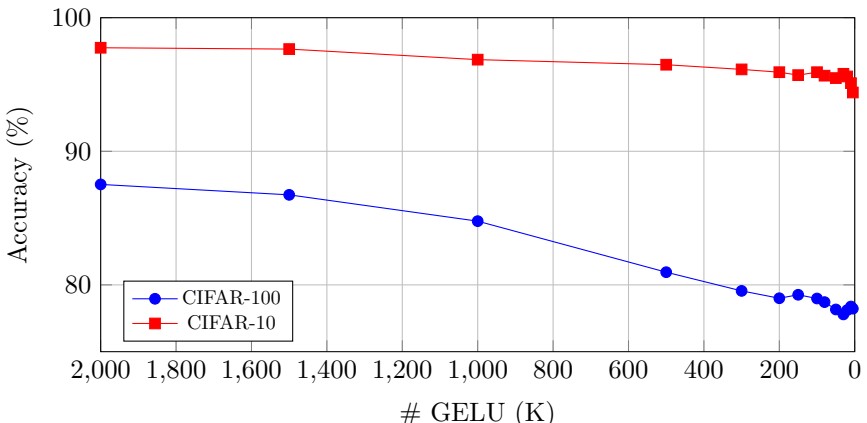

Figure 10: PriViT's ability to linearize GeLU operations visualized through performance on CIFAR datasets. As GELU operations decrease, CIFAR-100 and CIFAR-10 accuracies are affected, showcasing the trade-off between operation count and accuracy.

**Effect of using pre-trained checkpoints.** To further investigate why using pretrained checkpoint is improving performance, we report the non-linear distributions searched by PriViT and compare it with PriViT without pretrain for the nonlinearity budget of 315k and 320k respectively. We observe from our findings in figures 11,12 that the distribution found by the two methods differs across each layer. This supports our theory as to how PriViT operates under a strategic 'top-down' paradigm. Starting with a fine-tuned model, it has the advantage of an architecture that has not just discerned overarching generalization patterns but has also selectively pruned irrelevant information, streamlining its focus for a specific downstream task. This reduction of redundancy, undertaken from a vantage point of a pre-existing knowledge base, gives PriViT an edge.

## B  Supplementary graphics

The following figure shows a graphical representation of the switching operation.

**Search granularity.** An important characteristic of PriViT is it's flexibility to search over different granularity of non-linearities. GELU is a pointwise functions, thus PriViT can search either at embedding level or at a token level. On the other hand, softmax is a token level operation, thus it cannot be broken into a finer search space. Note that softmax operations can be extended to search over the head space or layer space, and similarly GELU can be searched over the layer space. Figure 14 illustrates the search granularity over token and embedding space.

**Parametric mask.** Figure 13 is an illustration of the working mechanism of the parametric mask introduced in PriViT. When the parameters are binarized it selects one of the two candidate function in the attention mechanism, and the gelu activation.

**PriViT overview.** Figure 15 provides an illustration of the complete PriViT algorithm, there are three distinct phases namely Finetuning the teacher, joint optimization of network and parametric masks, and final finetuning.

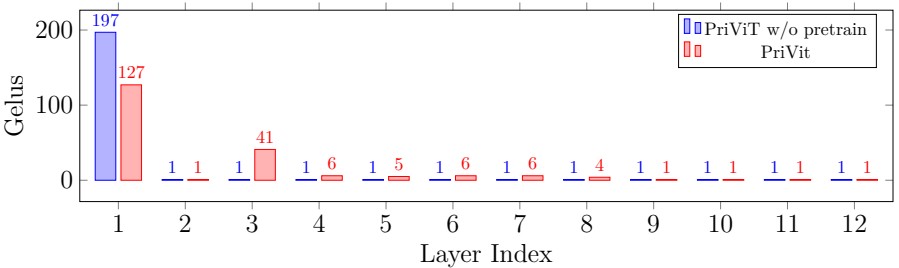

Figure 11: We compare the distribution of 208 GELU and 200 GELU operations distributed by PriViT w/o pretrain and PriViT respectively over tiny imagenet dataset.

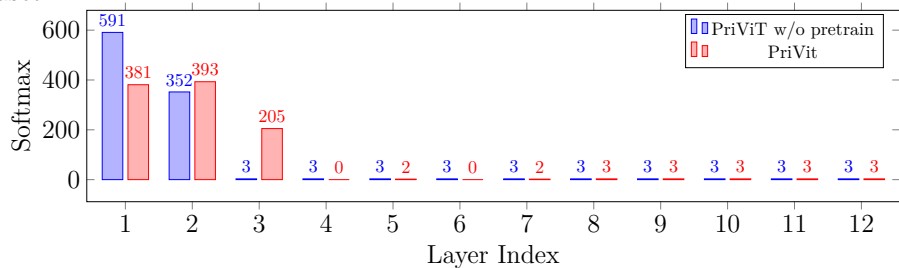

Figure 12: We compare the distribution of 973 softmax operations and 998 softmax operations operations distributed by PriViT w/o pretrain and PriViT respectively over tiny imagenet dataset.

## C  Role of Delphi in our framework

Delphi Mishra et al. (2020) encompasses two primary elements: a secure communication protocol and a NAS technique, specifically an evolutionary method. The secure communication aspect focuses on cryptographic protocols for neural networks within a multi-party computation setting. The NAS technique, known as Delphi's planner, is designed primarily to eliminate ReLU operations from neural architectures.

In our paper, we utilize Delphi's secure communication protocol for private inference. Furthermore, we took inspiration from Delphi's planner to design an effective method to reduce non-linear operations for attention-based architectures, particularly in Vision Transformers (ViTs). This enhancement was necessary because the original Delphi's planner is not optimally equipped for reducing non-linearities like softmax, gelu, and layernorms in ViT architectures.

Figure 16 provides an estimate of Delphi's planner lower bound on latency on ViT-Base with 80M parameters. To estimate this lower bound, we run PriViT on ViT-Base but only replace GELU operations with identity operations, till no Gelus were left to linearize, i.e. the left most point in the cyan curve indicates a network with no Gelus. All softmax and layernorms are left intact, Delphi's planner lower bound on latency is defined by the red vertical line which represents the latency of a model with no GELUs, and all softmax/layernorm intact. We compare this lower bound with the latency accuracy curve produced by PriViT on ViT-Base that Taylorize both softmax and GELUs.

## D  Attention Variants

Here we describe formally the different attention variant we ablated. Uniform form attention is basically described by the following equation

$$\text{UniformAttn}(\mathbf{X}) = \frac{(1)}{N}\mathbf{X}\mathbf{W}_v, \tag{7}$$

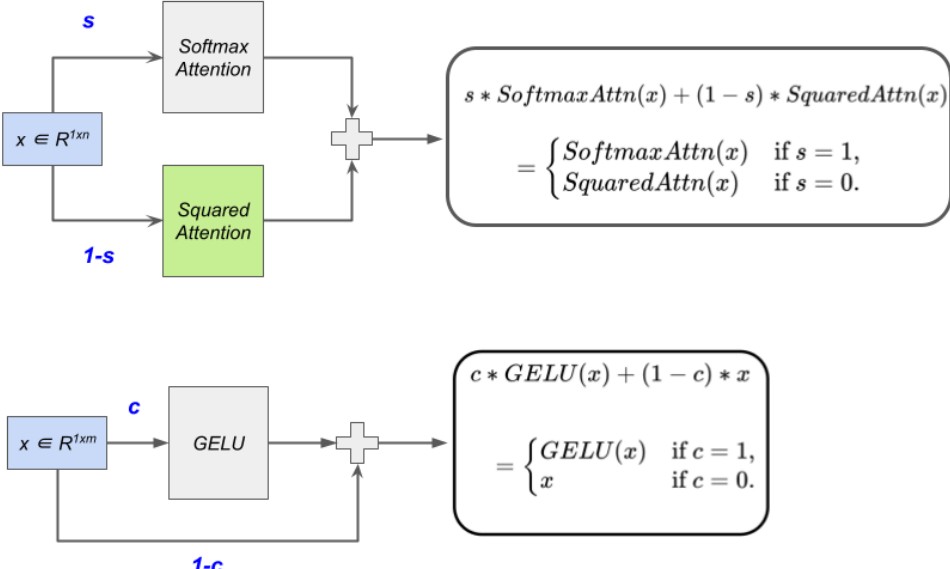

Figure 13: Parameterized Gelu and Self-Attention operations. **Top**: Tokens undergo softmax and squared attention in training. Post-training, parameter $S$ is frozen and binarized, selecting only one operation. **Bottom**: Embeddings pass through GeLU and Identity during training. Afterwards, parameter $C$ is frozen and binarized, choosing a single operation.

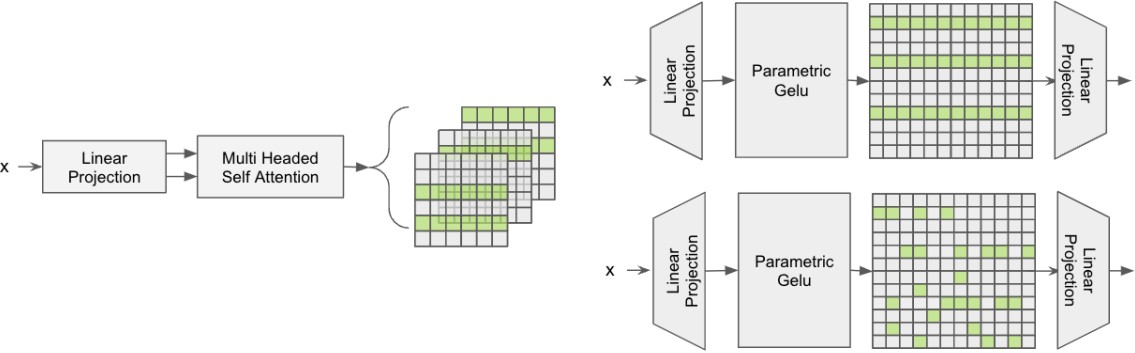

Figure 14: **Left**: The green blocks are SQUAREDATTENTION, and the grey blocks are Softmax Attention. For parametric attention, tokens emerge from a blend of softmax and square attention (refer to fig 13). Post-training, auxiliary variable $S$ is set to 0 or 1, resulting in $2^{N \times H}$ potential combinations per encoder block. **Right**: The green blocks are Identity function, and the grey blocks are GELU activation. Embeddings combine GELU and identity operations during training, as seen in fig 13. After training, parameter $C$ is frozen and binarized. This yields potential combinations of either $2^{H \times N}$ or $2^{N \times H \times m}$ for each ViT encoder block. Note that GELU being a pointwise function, we possess the flexibility to expand our search space either to tokens or directly to individual embeddings.

Where N is the number of tokens, so for each token the attention weights are equal hence the name UniformAttention.

ScaleAttn is the softmax candidate used in the work Zeng et al. (2022) which is essentially described as

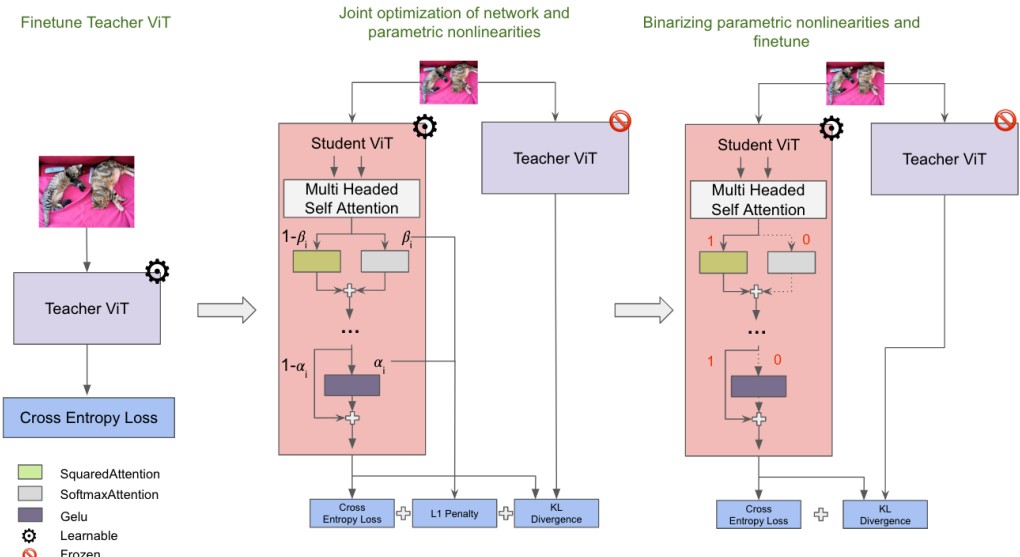

Figure 15: **Left**: Step 1 - Fine-tuning of a pretrained ViT over target dataset to produce the 'teacher ViT'. **Middle**: Step 2 - Duplicate teacher ViT, introduce parametric GELUs and attention mask to form 'student ViT'. Train using cross-entropy loss, KL divergence, and L1 penalty to gradually find a sparse mask. Binarize the mask post desired non-linearity budget. **Right**: Step 3 - With a frozen, binarized mask, further fine-tune the student model using cross-entropy loss and KL divergence with the teacher.

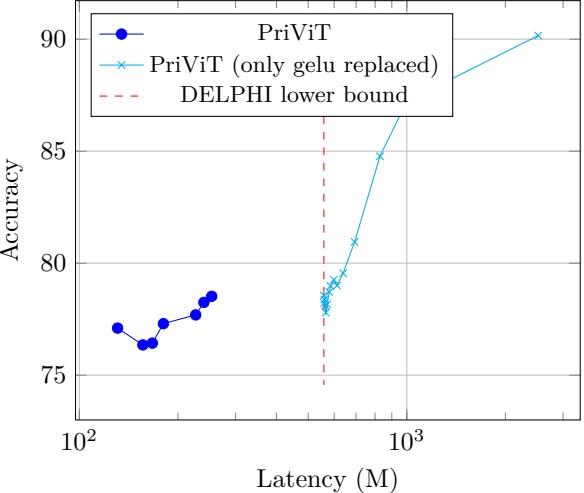

Figure 16: Estimation of Delphi's latency lower bound compared to PriViT. We use ViT-Base as our architecture and CIFAR 100 as our dataset.

$$\text{SCALEATTN}(\mathbf{X}) = \frac{\left(\mathbf{X}\mathbf{W}_q\mathbf{W}_k^T\mathbf{X}^T\right)}{N}\mathbf{X}\mathbf{W}_v, \tag{8}$$

