# OpenReview forum: "PriViT: Vision Transformers for Private Inference"
_TMLR — Accepted by TMLR_

### Review · Reviewer_S5rT · 2024-07-18

**Summary Of Contributions:**

The work aims to develop private inference for ViT's in an efficient, accuracy-preserving way. Unfortunately, ViT traditionally involves a lot of operations that are not amenable to standard private inference protocols such as Delphi. This work addresses this by taking a pretrained ViT and training it to switch out as many difficult operations (GeLU, Attention) for simpler alternatives (Identity, SquaredAttention) while still maintaining low error and distance to the original model. This method appears to achieve SOTA tradeoff between accuracy and inference latency. Other interesting findings include that many non-linearities are superfluous (since they can be replaced with simpler counterparts) and more important at the start (since these tend to be replaced less).

**Audience:**

Yes

**Claims And Evidence:**

Yes

**Requested Changes:**

All in weaknesses/questions above.

**Strengths And Weaknesses:**

### Strengths
- The work features many good ablations and corresponding analysis. I especially liked the following
  - The comparison between Squared Attention and other replacements for Attention
  - Comparing PriViT with/without pretraining, and with/without knowledge distillation
  - Analyzing the distribution of non-linearities replaced over layers
- The work provides a detailed comparison/contextualization to existing works at the start of the paper

### Weaknesses
- What are the differences between this work and Cho et al except for replacing Attention with Squared Attention? If I understand correctly, the algorithm in this work is the natural analogue of SNL for transformers. Therefore, the primary contribution comes from the analysis of the algorithm, not the algorithm itself.
- The baseline considered in this work is very conceptually similar to the method at hand. Does PriViT improve upon other approaches to private inference? Though other papers such as SAL-ViT and RNA-ViT haven't released their models, is it possible to compare against their numbers?

### Questions

- Is the citation for MPCViT on the first line of implication on page 3 meant to be Cho et al or Zeng et al?
- How much work was it to privatize your algorithm using Delphi? Did it make sense to consider any alternatives? I wonder if practitioners would be interested in reading about these difficulties.
- Is it possible to provide a plot demonstrating the Pareto frontier of accuracy vs inference latency for prior/current work? I understand the table already contains the information in numeric form.

---

> ### Author Response · Authors · 2024-08-11
> **Response to review**
>
> Thank you for your helpful comments. We are glad that you appreciated the many ablations and analysis we performed in our work.
> To respond to the listed weaknesses:
> * **Differences to SNL**: You are correct that algorithmically, PriVIT can be viewed as an extension of selective network linearization (SNL, Cho et al. ICML 2022) to the case of vision transformers. There are certain essential distinctions that arise due to the unique structure of ViTs. It is not enough to merely linearize certain nonlinearities in transformers; Figure 5 shows that switching to a quadratic approximation of attention at certain locations works better than straightforward linearization. Moreover, the presence of layernorms (which boil down to computing input-dependent inverse square roots)  in ViTs complicates both the choice of where to linearize, and how to properly account for latency costs.
> * **Comparison with RNA-ViT**: As described in our paper (Page 4), we deliberately avoided comparing our work with SAL-ViT and RNA-ViT. Both approaches assumed a hybrid ViT model (CCT), a different cryptographic security model (trusted third party), and did not make their code public for reproducibility purposes. Given these constraints, we opted to focus our comparisons on MPCViT instead.
>
>    Still, to address your comment, we considered performing quantitative comparisons with RNA-ViT. However, we found that RNA-ViT still has no checkpoints/code publicly available and the short response window made it infeasible to faithfully reproduce their models and training algorithms from scratch.
>
>    Therefore, in lieu of quantitative comparisons, we provide some qualitative reasoning for why we believe PriViT would perform better than RNA-ViT. There are two main algorithmic contributions by RNA-ViT: a compressed attention map, and an approximation for the GeLU. On a fundamental level, this is rather similar to MPCViT which performs softmax approximations (with an additional GeLU approximation in MPCViT+). The compressed attention map performed by RNA-ViT aims to shrink the attention map right before the softmax operation and then expand the attention map afterwards. By analogy to our results comparing PriViT to MPCViT and MPC-Tiny, we posit that the number of softmaxes being performed by PriViT is still significantly lower. A similar argument can be made for the GeLU approximation because PriViT enables fine-grained control of switching GeLUs on a scalar level.
>
>    We will be happy to make more detailed quantitative comparisons following a public release of RNA-ViT code.
>
> * **Citation Question**: We thank the reviewer for pointing out this error. We will fix this in the revised version.
>
> * **Difficulties with Delphi**: We chose Delphi purely as a matter of convenience for producing latency estimates. Similar protocols such as CryptFlow2 (Rathee et al, 2020) may equally be used, and we expect all trends to be similar (even though the precise numbers in plots/tables may change). Note that our focus in this paper is on designing architecture (specifically in the case of transformers) keeping the particular cryptographic protocol fixed. An interesting direction of future work may be co-design of architectures and protocols for supporting efficient private inference.
>
> * **Plot of Pareto Frontier**: Indeed, and we point the reviewer to Figure 3 in our manuscript. This is a plotted version of the latency/accuracy table of results.

---

> ### Comment · Reviewer_S5rT · 2024-08-19
>
> Thanks for responding to my concerns, though I still think better benchmarking/comparison would make the paper stronger. I maintain my current assessment reccomending acceptance.

---

> > ### Author Response · Authors · 2024-08-19
> > **Ack**
> >
> > Thank you for the careful review!

---

### Review · Reviewer_jc19 · 2024-07-29

**Summary Of Contributions:**

This paper introduces a method for adapting ViT-based model architectures for efficient and secure multi-party computation.

**Audience:**

Yes

**Claims And Evidence:**

No

**Requested Changes:**

- Critical
  - Please addresses weaknesses i.e. experiments on a dataset with larger dataset as well as fairer comparison to the baseline.

- Recommended:
  - Exploring the effectiveness of the method when scaling the ViT architecture (for ex. to small/base) would also help strengthen the paper.

- Presentation/Clarity:
  - The presentation of the main results, split across table 2 and 6 make it rather cumbersome for the reader to follow. It is recommended to consolidate them in one table.
  - The efficiency comparisons in table 4 are missing details; does the comparison account for the fact that PriViT employs pretrained checkpoints? Does MPCViT employ similar pretraining? Please provide details.

**Strengths And Weaknesses:**

Strengths:

- The paper exposition is clear, and the method analysis is easy to follow.
- The proposed method is simple and yet effective in outperforming existing baselines.

Weaknesses:
- Experiments: The presented comparisons are somewhat limited, and currently not enough to demonstrate the superiority of proposed approach. More specifically,
  - a) The considered datasets/settings are rather small in terms of image resolution (only up to $64 \times 64$), which naturally leaves a lot of redundancy in the image when upsampled and does not represent a standard use-case for ViTs. Thus, the authors need to demonstrate the method's effectiveness for datasets with larger and more diverse images (for example, Imagenet-1k)
  - b) In terms of the comparisons against MPCViT, I understand the reasoning behind comparing latencies with matched accuracies but given that MPCViT employs much smaller models which are likely to have lower accuracy ceilings, the comparison may be somewhat unfair. It would help the paper to provide a comparison to MPCViT used with architectures of a similar scale.

---

> ### Author Response · Authors · 2024-08-11
> **Response to review**
>
> Thank you for your helpful comments. We are glad that you appreciated the clarity of our writing, and analysis and the effectiveness of our method.
>
> In response to your listed weaknesses:
> * **Fair Comparison to MPCViT**: We appreciate the reviewer’s comments about the fairness of our comparisons to MPCViT. We request the reviewer to reference the general comment at the top regarding the new family of models (that we call MPC-Tiny) that presents a fairer analogue with MPCViT. Via these new experiments we show that PriViT performs significantly better in terms of private inference latency than MPC-Tiny in terms at similar iso-accuracies.
> * **Dataset Choice**: We agree that the current set of datasets used to benchmark PriViT does not represent real-world use cases of vision transformers. However, we would like to emphasize that private inference is a somewhat nascent field, and that all private inference works for computer vision have focused on precisely the three datasets (CIFAR-10/100, TinyImageNet) for which we report results.
>
>    In principle, we fully agree that methods such as PriViT and any future private inference works would benefit from benchmarks performed on more practical datasets. As a side note, we would like to note that PriViT natively scales up images to a resolution of $224 \times 224$ before passing the images to the ViT. Therefore, inference on PriViT will be natively achievable with datasets such as ImageNet.
>
> * **Exploring other ViT Sizes**: Indeed, we would like to point the reviewer to Section A of the Appendix, “PriViT on different size architectures”, where we address this point in detail.  In this section we explore the accuracy/latency tradeoff for ViT models of various sizes. In addition, in “PriViT on different ViT architectures” we explore the performance of PriViT when using the DeiT as a backbone instead of ViT-Tiny. We see similar accuracy/latency curves in both of these cases compared with the main PriViT results.

---

> > ### Comment · Reviewer_jc19 · 2024-08-25
> > **Response to Rebuttal**
> >
> > Thanks for responding to my concerns.
> >
> > - Regarding the dataset choice, I still do believe that pushing beyond the three mentioned datasets helps both in making the paper stronger and pushing the evaluation standards further. Specifically, even if the ViT scales natively and there are no natural comparative baselines available, understanding the limitations of the method when the images are natively at 224 X 224 and therefore inherently more complex, is still important.
> >
> > - Overall, I do appreciate the efforts of the authors towards implementing fairer comparisons on MPCViT and I am now inclined to recommend acceptance.

---

> > > ### Author Response · Authors · 2024-08-25
> > > **Ack**
> > >
> > > Thank you for your consideration!

---

### Review · Reviewer_fJqe · 2024-07-30

**Summary Of Contributions:**

Nowadays, because of user data privacy, the importance of private inference using secure multi-party computation protocols is increasing. However, most researchers have focused on CNN-based architectures because transformer-based architectures are ill-suited for private inference. Unlike CNN-based models, transformer-based models (e.g., ViT) use a large number of non-polynomial operations such as GELUs, softmaxes, and layer norms. The authors replace these nonlinear operations with Taylorization.

**Audience:**

Yes

**Broader Impact Concerns:**

It seems that there is no concern about ethical issues.

**Claims And Evidence:**

No

**Requested Changes:**

I think it would be better for the authors to supplement the comparison parts of the paper. In my opinion, the comparison with RNA-ViT (see W1) and using the same architecture when comparing the methods (see W2) are needed.

**Strengths And Weaknesses:**

S1 - PriViT utilizes a simple method to replace non-linear functions in ViTs. The methods are well-motivated and simple to adopt in other transformer-based architectures.

S2 - The proposed method shows high performance in both classification and accuracy.

W1 - I wonder why there is no comparison with RNA-ViT in the paper. Since RNA-ViT originates from the approximation of a softmax function, PriViT and RNA-ViT may be similar. It seems that the RNA-ViT method can be applied to the ViT structure as described in the RNA-ViT paper. It would be better for the authors to compare their method to RNA-ViT quantitatively and qualitatively.

W2 - The base models of PriViT and MPCViT are different; furthermore, they have different input image sizes and parameters. For a fair comparison, especially for accuracy, I think using the same architecture is necessary. If it is hard due to any method-related reasons, a deep analysis considering those conditions is needed.

---

> ### Author Response · Authors · 2024-08-11
> **Response to review**
>
> Thank you for your helpful comments. We are glad that you found our methods well motivated and adaptable to other transformer-based architectures.
>
> In response to the listed weaknesses:
> * **Fair Comparison to MPCViT**: We appreciate the reviewer’s comments about the fairness of our comparisons to MPCViT. We request the reviewer to reference the general comment at the top regarding the new family of models (that we call MPC-Tiny) that presents a fairer analogue with MPCViT. Via these new experiments we show that PriViT performs significantly better in terms of private inference latency than MPC-Tiny in terms at similar iso-accuracies.
> * **Comparison to RNA-ViT**: As described in our paper (Page 4), we deliberately avoided comparing our work with SAL-ViT and RNA-ViT. Both approaches assumed a hybrid ViT model (CCT), a different cryptographic security model (trusted third party), and did not make their code public for reproducibility purposes. Given these differences in approach, we opted to focus our comparisons on MPCViT instead.
>
>    Still, to address your comment, we considered performing quantitative comparisons with RNA-ViT. However, we found that RNA-ViT still has no checkpoints/code publicly available and the short response window made it infeasible to faithfully reproduce their models and training algorithms from scratch.
>
>    Therefore, in lieu of quantitative comparisons, we provide some qualitative reasoning for why we believe PriViT would perform better than RNA-ViT. There are two main algorithmic contributions by RNA-ViT: a compressed attention map, and an approximation for the GeLU. On a fundamental level, this is rather similar to MPCViT which performs softmax approximations (with an additional GeLU approximation in MPCViT+). The compressed attention map performed by RNA-ViT aims to shrink the attention map right before the softmax operation and then expand the attention map afterwards. By analogy to our results comparing PriViT to MPCViT and MPC-Tiny, we posit that the number of softmaxes being performed by PriViT is still significantly lower. A similar argument can be made for the GeLU approximation because PriViT enables fine-grained control of switching GeLUs on a scalar level.
> We will be happy to make more detailed quantitative comparisons following a public release of RNA-ViT code.

---

### Author Response · Authors · 2024-08-11
**General comment (1/3)**

**Fair comparisons with MPCViT**

The majority of our numerical comparisons are in relation to MPCViT, which is an important predecessor to our work. In our manuscript, we showed that PriViT achieves a better Pareto frontier for private inference than MPCViT by achieving lower latencies at iso-accuracies across various datasets.

However, the reviewers rightfully point out that the comparisons are not completely applies-to-apples due to (i) differences in model size, and (ii) differences in pre-training. This is because MPCViT uses a smaller transformer architecture than the standard ViT backbone (which we use), and we felt that focusing on standard ViT’s is the more relevant setting. Moreover, using a larger backbone should adversely affect private inference, but PriViT does better anyway (as we pointed out in Page 8). Finally, we were unable to find MPCViT implementations that fully replicated their results; our previous results were only based on trained checkpoints.

Nonetheless, we have tried our best to achieve a fair comparison with MPCViT over the last 2 weeks. We perform the following:
* We take the pre-trained ViT-Tiny checkpoint used in PriViT and apply NAS as described in the MPCViT paper.
* We perform 300 epochs of NAS for each dataset in our comparisons (CIFAR-10/100 and Tiny-ImageNet). This is in line with the number of epochs performed by the MPCViT authors reported in their paper.
* We then train this (post-NAS) ViT-Tiny on each benchmark dataset for varying values of μ. The μ parameter represents the percentage of attention heads that use ReLU softmax instead of Scaled-Attention.
* We train four models per dataset with μ values of 0.1, 0.3, 0.5, and 0.7 to match the training scheme of the MPCViT.
* We were not able to replicate the knowledge distillation results present in the MPCViT work. We used the code recently updated in the public MPCViT Github repository, but in our experiments found that the loss did not converge. Therefore, we opted to use a standard cross entropy loss function instead, and achieved reasonable accuracies with this method.
* We trained the models for 300 epochs to match the training time of the MPCViT work. This leads us to a family of 12 models (four for each dataset). We call this “MPC-Tiny”.

We present comparisons for PriViT with this new family of models in the table below. The benchmarking process is the same as the one presented in our paper. Note that there are slight changes in the times presented for PriViT and the times presented in our original paper. This is due to the benchmarking software, SecretFlow; the latency measurement using SecretFlow varies a bit based on the precise computing environment, but we see that order-of-magnitudes are similar for all tests.

---

> ### Author Response · Authors · 2024-08-11
> **Comment (2/3)**
>
> **New Comparison Tables**
>
> Table 1: MPC-Tiny (aka standard ViT backbone)
>
> | Mu  | Dataset        | Accuracy | Latency (seconds) |
> |-----|----------------|----------|-------------------|
> | 0.7 | Cifar10        | 91.48    | 42.03             |
> | 0.5 | Cifar10        | 88.48    | 40.28             |
> | 0.3 | Cifar10        | 93.97    | 36.80             |
> | 0.1 | Cifar10        | 94.49    | 34.21             |
> |     |                |          |                   |
> | 0.7 | Cifar100       | 40.22    | 43.17             |
> | 0.5 | Cifar100       | 79.12    | 39.37             |
> | 0.3 | Cifar100       | 75.04    | 36.82             |
> | 0.1 | Cifar100       | 76.21    | 34.47             |
> |     |                |          |                   |
> | 0.7 | Tiny Imagenet  | 66.66    | 42.98             |
> | 0.5 | Tiny Imagenet  | 65.46    | 39.30             |
> | 0.3 | Tiny Imagenet  | 67.57    | 37.50             |
> | 0.1 | Tiny Imagenet  | 65.11    | 34.03             |
>
> Table 2: MPCViT (aka a much smaller Transformer)
>
> | Mu  | Dataset        | Accuracy | Latency (seconds) |
> |-----|----------------|----------|-------------------|
> | 0.7 | Cifar10        | 94.30    | 7.03              |
> | 0.5 | Cifar10        | 94.20    | 6.66              |
> | 0.3 | Cifar10        | 94.10    | 6.38              |
> | 0.1 | Cifar10        | 93.60    | 6.06              |
> |     |                |          |                   |
> | 0.7 | Cifar100       | 77.80    | 6.97              |
> | 0.5 | Cifar100       | 76.90    | 6.62              |
> | 0.3 | Cifar100       | 76.90    | 6.42              |
> | 0.1 | Cifar100       | 76.40    | 6.06              |
> |     |                |          |                   |
> | 0.7 | Tiny Imagenet  | 62.55    | 69.64             |
> | 0.5 | Tiny Imagenet  | 63.70    | 55.45             |
> | 0.3 | Tiny Imagenet  | 63.36    | 40.86             |
> | 0.1 | Tiny Imagenet  | 62.62    | 26.76             |
>
> Table 3: PriViT (aka standard ViT backbone)
>
> | Dataset        | Accuracy | Latency (Seconds) |
> |----------------|----------|-------------------|
> | Cifar10        | 96.58    | 8.15              |
> | Cifar10        | 95.88    | 7.28              |
> | Cifar10        | 95.58    | 6.88              |
> | Cifar10        | 94.58    | 6.42              |
> | Cifar10        | 94.24    | 6.30              |
> | Cifar10        | 93.60    | 5.91              |
> | Cifar10        | 93.62    | 5.88              |
> |                |          |                   |
> | Cifar100       | 80.60    | 7.30              |
> | Cifar100       | 79.78    | 7.20              |
> | Cifar100       | 79.27    | 7.04              |
> | Cifar100       | 78.49    | 6.83              |
> |                |          |                   |
> | Tiny Imagenet  | 78.88    | 17.95             |
> | Tiny Imagenet  | 78.16    | 15.92             |
> | Tiny Imagenet  | 75.50    | 12.49             |
> | Tiny Imagenet  | 64.46    | 9.53              |

---

> ### Author Response · Authors · 2024-08-11
> **Comment (3/3)**
>
> **Analysis**
>
> We now analyze our new results, starting with latency comparisons. We first see that all of the models present in MPC-Tiny are significantly slower than their counterparts in PriViT. Notably, the accuracy for the MPC-Tiny models is worse than both PriViT models and MPCViT models on average. This is likely due to the knowledge distillation procedure not transferring well to larger models. We also notice that for each μ value, the latency is similar regardless of the dataset, which is as expected and provides a good sanity check. In addition, we see that MPC-Tiny models have a relatively inflexible lower bound for latency. The μ parameter controls how much latency one can save by removing ReLU Softmaxes; even after we have reached μ=0.1, the latency is still very significant due to the large model size. This is in contrast to PriViT where we control the nonlinearity Taylorization at a scalar level (instead of attention head level) which enables us to push the latency down to significantly lower levels.
>
> A final note: let us address why we haven’t included MPCViT+ included in our new comparisons. First, due to the short duration of the response window, we wanted to invest time and compute into making fair and accurate comparisons with the main model MPCViT. Second, the MPCViT paper did not provide any results or checkpoints for MPCViT+ for Tiny-ImageNet; just CIFAR-10/100.
>
> Third, in lieu of results for MPCViT+, let us provide some qualitative justification as to why it might not substantially improve over MPC-Tiny. In Table 6 we see that there is minimal impact on the Pareto frontier when comparing MPCViT and MPCViT+, with about a 1%-5% difference in runtime when comparing MPCViT and MPCViT+ at similar accuracies. Therefore, we expect
>
> In conclusion, we show that PriViT can be fairly compared to MPC-Tiny in the place of MPCViT. We also show that PriViT has significantly lower latency at similar accuracies compared to MPC-Tiny. We also discuss our insights into why this may be the case.We will add these results and discussion to the revised version of our manuscript.

---

### Author Response · Authors · 2024-08-11
**Response to reviewers**

We thank the 3 reviewers and the AE for their consideration of our paper. The reviews were insightful and helped considerably strengthen the manuscript. If the reviewers/AE have any further comments, we are happy to continue the discussion. Thanks again.

---

### Decision · Action_Editor_2U14 · 2024-09-06

**Recommendation:** Accept as is

**Comment:**

The authors are recommended to consider the reviewer comments and thoroughly address them in the camera-ready version. Having said that, I do believe the paper passes the bar of the TMLR acceptance.

**Audience:**

Privacy-preserving models are an integral part of ML models; indeed their significance is expected to increase the coming years. This paper is relevant to the ML audience.

**Claims And Evidence:**

This paper modifies ViT for privacy-preserving applications. The idea is to replace operations that are not amenable to privacy settings (such as GeLU) with operations that are amenable to secure multi-party communication settings. The modified model maintains a good trade-off between accuracy and inference latency. The paper is clear and the proposed model is intuitive.